# Microbial Signatures in COVID-19: Distinguishing Mild and Severe Disease via Gut Microbiota

**DOI:** 10.3390/biomedicines12050996

**Published:** 2024-05-01

**Authors:** Julia S. Galeeva, Dmitry E. Fedorov, Elizaveta V. Starikova, Alexander I. Manolov, Alexander V. Pavlenko, Oksana V. Selezneva, Ksenia M. Klimina, Vladimir A. Veselovsky, Maxim D. Morozov, Oleg O. Yanushevich, Natella I. Krikheli, Oleg V. Levchenko, Dmitry N. Andreev, Filipp S. Sokolov, Aleksey K. Fomenko, Mikhail K. Devkota, Nikolai G. Andreev, Andrey V. Zaborovskiy, Petr A. Bely, Sergei V. Tsaregorodtsev, Vladimir V. Evdokimov, Igor V. Maev, Vadim M. Govorun, Elena N. Ilina

**Affiliations:** 1Research Institute for Systems Biology and Medicine, Department of Mathematical Biology and Bioinformatics, Moscow 117246, Russia; fedorov.de@gmail.com (D.E.F.); hed.robin@gmail.com (E.V.S.); paraslonic@gmail.com (A.I.M.); pavav@mail.ru (A.V.P.); 2Lopukhin Federal Research and Clinical Center of Physical-Chemical Medicine of Federal Medical Biological Agency, Moscow 119435, Russia; ks-selezneva@yandex.ru (O.V.S.); ppp843@yandex.ru (K.M.K.); djdf26@gmail.com (V.A.V.); maxim_d_morozov@mail.ru (M.D.M.); 3Department of Clinical Dentistry, Moscow State University of Medicine and Dentistry, Moscow 127473, Russia; olegyanushevich@yandex.ru (O.O.Y.); krikheli_msmsu@mail.ru (N.I.K.); truovl@gmail.com (O.V.L.); dna-mit8@mail.ru (D.N.A.); phlppsokolov@gmail.com (F.S.S.); docfomenko@yandex.ru (A.K.F.); devkota@bk.ru (M.K.D.); nicdoctor2009@yandex.ru (N.G.A.); azabor@mail.ru (A.V.Z.); pbely@ncpharm.ru (P.A.B.); sergiotsar@yandex.ru (S.V.T.); vvevdokimov@rambler.ru (V.V.E.); igormaev@rambler.ru (I.V.M.); vgovorun@yandex.ru (V.M.G.)

**Keywords:** COVID-19 severity, 16S, human gut microbiome, feces, SARS-CoV-2

## Abstract

The COVID-19 pandemic, caused by the SARS-CoV-2 virus, has significantly impacted global healthcare, underscoring the importance of exploring the virus’s effects on infected individuals beyond treatments and vaccines. Notably, recent findings suggest that SARS-CoV-2 can infect the gut, thereby altering the gut microbiota. This study aimed to analyze the gut microbiota composition differences between COVID-19 patients experiencing mild and severe symptoms. We conducted 16S rRNA metagenomic sequencing on fecal samples from 49 mild and 43 severe COVID-19 cases upon hospital admission. Our analysis identified a differential abundance of specific bacterial species associated with the severity of the disease. Severely affected patients showed an association with *Enterococcus faecium*, *Akkermansia muciniphila*, and others, while milder cases were linked to *Faecalibacterium prausnitzii*, *Alistipes putredinis*, *Blautia faecis*, and additional species. Furthermore, a network analysis using SPIEC-EASI indicated keystone taxa and highlighted structural differences in bacterial connectivity, with a notable disruption in the severe group. Our study highlights the diverse impacts of SARS-CoV-2 on the gut microbiome among both mild and severe COVID-19 patients, showcasing a spectrum of microbial responses to the virus. Importantly, these findings align, to some extent, with observations from other studies on COVID-19 gut microbiomes, despite variations in methodologies. The findings from this study, based on retrospective data, establish a foundation for future prospective research to confirm the role of the gut microbiome as a predictive biomarker for the severity of COVID-19.

## 1. Introduction

The COVID-19 pandemic, caused by the highly contagious SARS-CoV-2 virus, was officially declared in March 2020 and lasted for three years until May 2023. In three years, the SARS-CoV-2 virus claimed millions of lives worldwide, becoming one of the deadliest viral diseases in decades [1].

Apart from respiratory symptoms, the SARS-CoV-2 virus is now recognized to have extrapulmonary effects on different organs and systems. Gastrointestinal symptoms are frequently observed, and emerging evidence suggests that COVID-19 patients undergo changes in their gut microbiome, encompassing bacteria, fungi, and viruses, persisting even in cases of long-term morbidity and post-COVID-19 syndrome [2,3,4]. These alterations disrupt the ecological network, diminish diversity, and disturb homeostasis within the gut. Concurrent comorbidities compromise immune states, and individual factors further contribute to the overall disease severity [5].

SARS-CoV-2, the virus responsible for COVID-19, enters human intestinal cells through the ACE2 receptor, triggering an inflammatory response involving cytokines [6,7]. This infection can disrupt the mechanical barrier of the gastrointestinal tract by affecting the expression and function of tight junction (TJ) proteins, compromising the integrity of the intestinal paracellular barrier [8,9]. Recent research in mice has directly proved that SARS-CoV-2 infection leads to dysbiosis in the gut microbiome, characterized by changes in Paneth cells, goblet cells, and markers associated with barrier permeability [10]. Additionally, the analysis of blood culture results, along with microbiome data, suggests the potential translocation of gut bacteria into the systemic circulation of COVID-19 patients [10].

Numerous investigations have assessed the disparities in gut microbiota composition between SARS-CoV-2-infected patients and healthy individuals, as well as among patients with varying disease severities [11,12,13]. In a study from Germany, the researchers examined 108 COVID-19, 22 post-COVID-19 patients, 20 pneumonia controls, and 26 asymptomatic controls by using 16S rRNA gene sequencing [14]. They divided SARS-CoV-2-infected patients into three groups (mild/severe/death) based on the World Health Organization (WHO) Ordinal scale for clinical improvement. A differential analysis between study groups showed that zOTUs (zero-radiation operational-taxonomic units) belonged to *Faecalibacterium prausnitzii*, *Blautia luti*, *Dorea longicatena*, *Gemmiger formicilis*, and *Alistipes putredinis*, and were significantly decreased in severe, death, and post-COVID-19 groups; zOTUs belonging to *Clostridium innocuum*, *Ruthenibacterium lactatiformans*, and *Alistipes finegoldii* were increased in patients with a severe disease progression.

In another study [15], the researchers conducted a comparative analysis of the 16S gut microbiome composition among 30 COVID-19 patients, 24 influenza A (H1N1) patients, and 30 matched healthy controls. In the COVID-19 group, the dominant genera in the gut microbiome were *Streptococcus*, *Rothia*, *Veillonella*, *Erysipelatoclostridium*, and *Actinomyces*; the gut microbiome of the healthy control group was primarily dominated by the genera *Romboutsia*, *Faecalibacterium*, *Fusicatenibacter*, and the *Eubacterium hallii* group. Compared to the H1N1 group, the abundance of *Prevotella*, *Ezakiella*, *Murdochiella*, and *Porphyromonas* was higher in the H1N1 group than in the COVID-19 group. 

In our study, we investigated the composition of the gut microbiome in 92 inpatients with COVID-19 by using 16S rRNA gene sequencing of the V3–V4 region. To assess the severity of the disease, we utilized lung computed tomography (CT) scans to evaluate the degree of lung damage, and the WHO Performance Scale to assess the functional level of each individual. These assessments enabled us to classify the patients into two comparison groups based on the severity of their disease course.

Our objective was to identify microorganisms that are associated with the progression of the disease by performing differential abundance testing while considering patients’ metadata as covariates. Furthermore, we examined the variations in the structure of the microbial interaction network between these two groups.

We anticipate that the findings from our study will enhance the understanding of the mechanisms underlying the impact of clinical manifestations of COVID-19 on the human gut microbiota.

## 2. Materials and Methods

### 2.1. Samples and Data Collection

The study included 92 patients with laboratory-confirmed novel coronavirus infection SARS-CoV-2 and clinical manifestations of COVID-19, who were treated at the Kuskovo Multiprofile Clinical Medical Center, A.I. Evdokimov Moscow State Medical University from 15 April to 27 June 2021. Fecal samples were obtained from 49 patients with mild cases of COVID-19 and 43 patients with severe cases of COVID-19 upon their admission to the hospital. All the patients signed statements of informed consent to participate in the study. The study did not include patients diagnosed with cancer. 

Patients fulfilled all the following inclusion criteria:Men and women aged 18 years and older with a confirmed diagnosis of “U07.2 Coronavirus infection COVID-19 caused by SARS-CoV-2 virus” according to PCR (oropharyngeal-/nasopharyngeal swab);Mild and severe dynamics of the course of coronavirus infection (1–3 degrees of pneumonia severity according to CT scan);Signed voluntary informed consent for participation in the study.A patient was not included or excluded from the study if the subject met at least one of the following inclusion/exclusion criteria:Malignant neoplasms of any site;Chronic liver disease of infectious and non-infectious etiology;Chronic kidney disease;Chronic hepatic and renal insufficiency;Systemic and autoimmune diseases;Concomitant serious somatic and psychiatric pathologies;Pregnancy;Participation in another clinical trial currently or in the last 30 days;Any other medical or non-medical reason that, in the opinion of the investigator, may prevent the patient from participating in the trial.

The following metadata were collected based on questionnaires: whether or not the patient has health disorders such as hypertension, diabetes, obesity, and coronary artery disease (CAD). 

All patients signed an informed consent form to participate in the study and the study was approved by the Independent Interdisciplinary Committee for Ethical Review of Clinical Trials, protocol 01–21 of 28 January 2021.

Stool samples were collected in sterile containers in a sample volume of 5 to 15 mL. The samples were stored at −70 °C.

### 2.2. Study Cohort

The final group included samples from 92 patients, ranging in age from 20 to 88 years old, with a distribution of 48 females and 44 males. After admission to the hospital, each patient underwent a lung computed tomography (CT) scan to assess the degree of lung damage. The CT-based severity classification system provides an assessment of the extent of COVID-associated lung abnormalities observed on CT scans. The classification ranges from CT1 (up to 25% involvement) to CT2 (up to 50% involvement), CT3 (up to 75% involvement), and CT4 (up to 100% involvement) [16]. Among subjects, 43 (43/92, 46.7%) possessed a CT1 severity score, 40 (40/92, 43.5%) possessed a CT2 severity score, and 9 (3/92, 9.7%) possessed a CT3 severity score. 

In addition to the CT score, the patients’ condition at the time of hospitalization was evaluated using the WHO Performance Scale (WHO-PS) [17] according to the hospital’s internal regulations. This scale measures the patient’s functional level in terms of their ability to engage in self-care, daily activities, and physical capabilities. The scale comprises five levels of assessment:

0: The patient is fully active and can perform all tasks as before the disease.

1: The patient cannot perform heavy work but can handle light or sedentary tasks (e.g., light housework or clerical work).

2: The patient receives outpatient treatment, can take care of themselves, but is unable to work, and spends over 50% of their waking time in an upright position.

3: The patient can only manage limited self-care, spends time in a chair or bed, and more than 50% of their waking time in this state.

4: The patient is disabled, completely reliant on others for care, and confined to an armchair or bed.

5: The patient has passed away.

Based on this classification, out of the 92 subjects, 6 (6/92, 6.52%) had a WHO-PS level of 0, 7 (7/92, 7.6%) had a WHO-PS level of 1, 7 (7/92, 7.6%) had a WHO-PS level of 2, 57 (57/92, 61.9%) had a WHO-PS level of 3, and 15 (15/92, 16.3%) had a WHO-PS level of 4.

To facilitate further comparative analysis based on CT and WHO-PS scores, the 92 patients were categorized into two groups. The first group, labeled as “mild” (*n* = 49), consisted of patients with a mild disease course (CT1 + WHO-PS 0–3; CT2 + WHO-PS 0–2; CT3 + WHO-PS 0–2). The second group, labeled as “severe” (*n* = 43), comprised patients with a severe disease course (CT2 + WHO-PS 3–4; CT1 + WHO-PS 4; CT3 + WHO-PS 3–4) (Figure 1A).

Information regarding antibiotic consumption before hospitalization was acquired through patient self-reporting.

We analyzed the distribution of some comorbidities in the groups of patients with mild and severe COVID-19. We did not observe any statistically significant differences in the distribution of covariates between groups (Figure 1B–H). Patient characteristics are presented in Appendix A.

### 2.3. 16S rRNA Sequencing

The total DNA was isolated from 1 mg of each collected sample using the MagMAX DNA Multi-Sample Ultra 2.0 Kit (Roche) and a KingFisher Flex automated isolation station. Genomic libraries were prepared and sequenced according to the 16S metagenomic sequencing library preparation protocol for MiSeq, Illumina. Each sample was processed using the Tersus PCR kit (Evrogen, Russia); primers 341F and 801R were used to amplify the V3–V4 region of the 16S rRNA gene, and unique combinations of indexing primers, which were analogs of the primers from the Nextera XT Index kit v2, were used to barcode samples.

The sequencing procedure was performed on a HiSeq 2500 instrument using the HiSeq Rapid SBS Kit v2 (500) consumable kit (Illumina) according to the manufacturer’s recommendations

### 2.4. 16S rRNA Data Processing

Leftover adapters were removed using Trimmomatic v0.36 [18], and quality filtering of reads was performed with the *filterAndTrim* function from the DADA2 package [19]. Denoising, merging, and chimera removal was carried out with DADA2 v1.24.0 software with the following parameters: learnErrors: nbases = 1 × 10^9^, randomize = TRUE, MAX_CONSIST = 2, dada: pool = TRUE, mergePairs: minOverlap = 18, removeBimeraDenovo: allowOneOff = FALSE, method = “consensus”. 

Taxonomy annotation was carried out against the SILVA v138 reference database [20].

Potential contaminants were removed with the “frequency” method using the package *decontam* [21] version 1.10.0. In total, the 1600 samples were decontaminated (265 samples from this project and 1335 samples from other projects with similar objects for analysis) to better identify contaminant sequences.

The resulting dataset contained 6235 ASVs, with 2744 ASVs for the target dataset (*n* = 265). The mean number of reads per sample was 48,126. Samples with <1000 reads were removed. After the removal of missing metadata, 92 out of the original 265 samples were included in the study. The scheme of analysis is presented in Figure 2.

### 2.5. Statistical Analysis

The statistical analysis of the microbiome data was conducted using R version 4.0.5. The vegan package [22] and phyloseq package [23] were employed for the analysis. To filter out low-represented ASVs (Amplicon Sequence Variants), the *core_members* function from the microbiome package was applied with the criteria of detection threshold set at 15 and prevalence threshold at 10%. This filtering process resulted in 313 ASVs for further analysis.

The diversity composition of the bacterial microbiome was assessed using α-diversity, specifically the Shannon index, utilizing the *plot_richness* function from the phyloseq package.

Beta diversity was calculated with the principal coordinate analysis (PCoA) Unifrac weighted metric by using function *dist* from the vegan package and *plot_ordination* function from the phyloseq package. The Adonis test was used to determine statistical differences between the two groups.

The Dirichlet multinomial mixture (DMM) approach on a genus-level matrix of samples from all participants was performed to stratify the joint microbiome dataset into community types [24]. 

The permutational multivariate variance analysis (PERMANOVA) was performed to identify associations between taxa and host parameters. The Adonis function from the vegan package was used, running PERMANOVA with 1000 permutation tests on weighted UniFrac distance.

For differential abundance analysis at the ASV level, the DESeq2 package [25] in R and Songbird utility [26] in Python were used. The same formula was used for both instruments: ~group + age_group + sex + diabetes + obesity + hypertension + coronary artery disease + antibiotics, where the variable of interest was “group” (COVID-19 severity), and the other variables were used as covariates. In DESeq2, the default Wald test with Benjamin–Hochberg *p*-value correction was used to assess the statistical significance of log2FoldChange. A significance cut-off *p*-value < 0.05 and Log2FoldChange ≥ 1.5 were applied. Songbird was also utilized to determine differential rankings of microbes between the mild and severe groups. The overlapping set of differentially represented ASVs identified with both DESeq2 and Songbird was selected for further analysis.

PICRUSt2 v2.5.1 tool was used to reconstruct the phylogenetic tree [27].

The co-abundance networks of microbial ASVs were generated using the SPIEC-EASI [28] algorithm with the Meinshausen–Bühlmann method for detecting correlations from the NetCoMi package [29]. The following parameters were utilized: 100 subsamples, 25 lambda iterations, and a minimum lambda value of 0.001. From the resulting network, clusters of co-abundant ASVs were identified using the Louvain method. Eigenvector centrality and degree centrality were used for defining hubs/keystone taxa (nodes with a centrality value above the empirical 95% quantile).

## 3. Results

### 3.1. Taxonomy of the Gut Microbiome of Patients Infected with SARS-CoV-2

The resulting dataset consists of 313 ASVs that belong to 34 families, with an average of 36276 read counts per sample. Among the microbial composition of samples collected from 92 patients with varying degrees of COVID-19 severity, the top seven dominant families were identified as *Lachnospiraceae*, *Ruminococcaceae*, *Bacteroidaceae*, *Streptococcaceae*, *Coriobacteriaceae*, *Akkermansiaceae*, and *Oscillospiraceae* (Figure 3, Appendix A). The taxonomic profile at the ASV level is shown in Appendix A.

### 3.2. Alpha and Beta Diversity Shows No Significant Differences between Patients with Mild and Severe COVID-19

Microbial alpha diversity, assessed using the Shannon metric, was utilized to examine changes in the microbiota community structure among patients categorized into mild and severe groups. Figure 4A displays the outcomes, indicating that no statistically significant variations (Wilcoxon test) were observed in the alpha diversity indices between the two patient groups. Beta-diversity was calculated through a principal coordinate analysis (PCoA) by using the weighted Unifrac metric and showed no statistically significant variations (Adonis test) between the two severity groups, as shown in Figure 4B. 

### 3.3. Determining Gut Biotope Community Types in Mild and Severe COVID-19 Patients

We employed the Dirichlet multinomial mixture method to analyze a dataset comprising 134 genera at the genus level. This analysis revealed the presence of two distinct microbiota community types based on taxonomic composition (Appendix AA). Community type 1 consisted of 62 patients (mild = 31, severe = 31), while community type 2 included 30 patients (mild = 18, severe = 12). The dissimilarity of community types 1 and 2 is shown in Figure 5.

At the ASV level, the main drivers distinguishing type 1 were ASV64_Blautia, ASV124_Bacteroides, ASV6_Streptococcus, ASV74_[Ruminococcus] torques group, and ASV138_Subdoligranulum. Conversely, type 2 was characterized by ASV64_Blautia, ASV124_Bacteroides, ASV117_Faecalibacterium, ASV138_Subdoligranulum, and ASV114_Ruminococcus (Figure 6).

Additionally, our analysis found no significant differences in the distribution of age, sex, severity group, obesity, and diabetes between the two community types (Appendix AB–I). This implies that these factors do not play a significant role in determining the division of a taxonomic composition into these clusters.

### 3.4. Exploring the Relationship between the Gut Microbiome of SARS-CoV-2-Infected Patients and Covariates

For testing the association between the microbiome and such covariates as age, sex, severity of COVID-19 (group), hypertension, diabetes, obesity, and coronary artery disease (CAD), we applied a PERMANOVA (Figure 7). 

These covariates explained 17% of the microbiome taxonomic composition with the most extensive and statistically significant contribution coming from the comparison group (4%), antibiotics (3%), diabetes (2%), and obesity (2%).

### 3.5. Analyzing Microbial Associations with Disease Severity in Relation to Age, Gender, and Comorbidities

In our research focused on exploring the potential impact of gut microbiome composition on the severity of COVID-19, we considered several important metadata variables, including the patient’s age, sex, antibiotics intake, and comorbidities. Our objective was to identify ASVs that exhibited differential abundance patterns between patients with mild (*n* = 49) and more severe (*n* = 43) disease manifestations. To achieve this, we employed two distinct analysis methods: DESeq2 and Songbird.

During the DESeq differential abundance analysis, we observed significant differences in the abundance of 66 ASVs. Out of these, 42 ASVs were specifically associated with the mild degree group, while 24 ASVs exhibited associations with the severe degree group in the context of COVID-19 (Appendix A). To further explore the findings, we used the Songbird tool and selected an equal number of ASVs (mild = 42, severe = 24) from each group derived from DESeq2 (Appendix A). Then, we selected ASVs associated with both the mild and severe course of COVID-19 that overlap the results of the two instruments (elected an equal number of ASVs (mild = 13, severe = 20) from each group derived from DESeq2 (Appendix A, Appendix A). The different ASVs selected by both tools (*n* = 4) that aligned to the same species were also added to the analysis (Appendix A). 

Our analysis revealed a significant enrichment of 13 specific ASVs associated with the severe course of the disease. These ASVs originate from genera such as *Enterococcus*, *Rothia*, *Akkermansia*, *Agathobacter*, *Actinomyces*, *Alloprevotella*, *Slackia*, *Prevotella9*, [*Ruminococcus*] *gauvreauii* group, *Eubacterium*, *Eisenbergiella*, and *Christensenellaceae R-7* group. Upon performing a BLASTN alignment of the ASV sequences against the nt database, we identified the corresponding matches for each ASV as follows: ASV29 corresponds to *Enterococcus hirae*/*Enterococcus faecium*, ASV55 to *Rothia mucilaginosa*, ASV60 to *Akkermansia muciniphila*, ASV62 to [*Eubacterium*] *rectale*, ASV134 to *Schaalia odontolytica*, ASV822 from the *Alloprevotella* genus to an uncultured bacterium, ASV938 to *Slackia isoflavoniconvertens*, ASV1016 to *Prevotella copri*, ASV1444 from the [*Ruminococcus*] *gauvreauii* group genus to a *Blautia fasciola* (identity = 99.5%)/*Lachnospiraceae bacterium* (identity = 99.7%), ASV1564 to *Eubacterium limosum*, ASV1629 to *Enterococcus faecium*, ASV1650 to *Eisenbergiella tayi*, and ASV2021 from *Christensenellaceae R-7* group as an uncultured bacterium.

We found 24 ASVs from Veillonella, Haemophilus, Alistipes, Atopobium, CAG-352 Bacteroides, Streptococcus, Romboutsia, Prevotella_9, Anaerostipes, [Eubacterium] xylanophilum group, Paraprevotella, Blautia, Turicibacter, [Eubacterium] hallii group, [Eubacterium] ventriosum group, Faecalibacterium genera that were associated with the mild group. ASV4 has been identified as Veillonella dispar/Veillonella nakazawae, ASV14 as Haemophilus parainfluenzae, ASV35 aligns with an uncultured organism (identity = 100%)/Atopobium parvulum (identity = 99.5%), ASV50 as Ruminococcoidesnbili/Ruminococcus sp, ASV52 as Streptococcus gordonii, ASV177 Romboutsia ilealis, ASV202 as Alistipes putredinis, ASV284 as Bacteroides vulgatus, ASV277 and ASV347 both aligned with Prevotella copri, ASV545 as Lachnospiraceae bacterium sunii, ASV773 as Bacteroides stercoris, ASV1277 from the Bacteroides genus as *Phocaeicola* sp. (identity = 100%)/Bacteroides massiliensis (identity = 99.7%), ASV1326 as Blautia faecis, ASV177 as Anaerostipes hadrus, ASV1797 as Eubacterium sp., ASV1827 as Anaerostipes hadrus, ASV1925 from the Paraprevotella genus as an uncultured bacterium, ASV2019 as Turicibacter sanguinis, and ASV2362 as Blautia sp.. Among different ASVs selected by both tools, ASV1870 and ASV1360 were identified as Anaerobutyricum hallii, ASV1494 and ASV556 as Eubacterium ventriosum, ASV707 and ASV998 as Faecalibacterium prausnitzii, and ASV949 and ASV1067 as Prevotella copri.

Figure 8 shows all differentially abundant ASVs associated with the COVID-19 severity course analyzed by the Deseq2 and Songbird utilities.

### 3.6. The Gut Microbiota’s Network Structure Shows Variations across COVID-19 Patients with Differing Severity Levels

We utilized SPIEC-EASI to investigate bacterial interactions and evaluate potential variations in the organization of microbial communities between the mild and severe groups. This analysis focused on 313 highly represented ASVs in the dataset, aiming to gain insights into the microbial dynamics and community structure associated with different disease severity levels. The network analysis revealed distinct characteristics in the microbial communities of the mild and severe groups (Appendix A, Appendix A). The network of the mild group exhibited a higher total number of components (mild = 169, severe = 131), higher clustering coefficient (mild = 0.155, severe = 0.038), higher edge density (mild = 0.097, severe = 0.025), and natural connectivity (mild= 0.051, severe = 0.013) compared to the severe group. However, the relative size of the largest connected component (LCC) (mild = 0.080, severe = 0.27), shorter average path length (mild = 2.63, severe = 4.8), and modularity value (mild = 0.53, severe = 0.75) were smaller in the mild group. 

The network analysis identified four hubs/keystone taxa for the mild group, including ASV432_Dorea, ASV372_Blautia, ASV387_Coprococcus, ASV655_UCG005, and three hubs/keystone taxa for the severe group: ASV2222_Lachnospiraceae FCS020 group, ASV1365_Lachnospiraceae FCS020 group, and ASV2294_Marvinbryantia. After conducting a BLASTN alignment of the ASV sequences against the nt database, we matched each ASV with its respective counterpart as follows: ASV432_Dorea as *Dorea formicigenerans*, ASV372_Blautia as *Blautia obeum*, ASV387_Coprococcus as *Coprococcus comes*, ASV655_UCG005 as an uncultured bacterium, ASV2222_Lachnospiraceae FCS020 group as *Lachnospiraceae bacterium*, ASV1365_Lachnospiraceae FCS020 group as *Lachnospiraceae bacterium*, and ASV2294_Marvinbryantia as *Jingyaoa shaoxingensis*/uncultured bacterium, respectively (Appendix A).

## 4. Discussion

In our study, we utilized the 16S rRNA metagenomic analysis to compare the taxonomic composition of the gut microbiota in COVID-19 patients upon admission to the hospital. The clinical samples from this investigation were processed and sequenced in a way that was similar to those from other COVID-19-related projects conducted in our lab. The results of the upper respiratory sample analysis have been published [30].

The actual study focused on the analysis of fecal samples from COVID-19 patients with complete metadata, including age, sex, CT lung scans, WHO-PS, antibiotic usage, and comorbidities such as hypertension, coronary artery disease, diabetes, and obesity. By combining WHO-PS scores and CT scans, we categorized patients according to disease severity. To our delight, the selected severe (*n* = 43) and mild (*n* = 49) groups showed no differences in the distribution of age, sex, antibiotic usage, hypertension, coronary artery disease, diabetes, and obesity. Although we did not observe significant differences between COVID-19 severity groups to these comorbidities, understanding risk factors for contracting SARS-CoV-2 remains crucial for managing COVID-19 outcomes. Identifying risk factors such as hypertension, obesity, inflammatory bowel disease, and other risk factors, enables the tailoring of preventive measures and treatments, potentially mitigating the severity and spread of the disease [31,32]. Furthermore, risk factor analysis is essential for identifying vulnerable populations, allowing healthcare providers to optimize resource allocation and enhance patient care strategies during the pandemic [33].

This paper compares individuals with mild COVID-19 cases to those with more severe cases, rather than comparing healthy individuals to COVID-19-infected subjects. The decision to concentrate on disease severity is driven by the fact that the comparison of healthy individuals and COVID-19-infected individuals has been extensively studied in the literature. Moreover, the Anna Karenina principle, which highlights that the gut microbiome of healthy individuals tends to be less variable and more stable than that of individuals with various diseases [34], also supports our choice. Additionally, considering that SARS-CoV-2 can be present in healthy individuals who may be asymptomatic carriers, using asymptomatic individuals as a control group, especially during a pandemic, may introduce bias into the analyses. Therefore, we have opted to examine the gut microbiota of patients with different degrees of disease severity caused by clinical manifestations of SARS-CoV-2.

The taxonomic composition of fecal samples from COVID-19 patients aligns with previous studies on human gut microbiota [35,36]. Our samples exhibited a homogeneous taxonomic profile, as evident in the heatmaps of Appendix A, where we observed no apparent clustering based on the considered covariates.

We found no statistically significant differences in alpha and beta diversity in the gut microbiota of mild and severe COVID-19 patients. Our results differ from others that have compared patients with different COVID-19 severity levels [14,37]. Various factors can contribute to these inconsistencies, such as variations in study design, sample size, patient demographics, methodologies, and geographic locations. Furthermore, there is no standardized or consistent method to categorize patients into severity groups. The lack of a uniform approach can contribute to variations in results across different studies and highlight the challenges in accurately assessing and comparing COVID-19 severity.

We observed significant differences in the abundance of specific ASVs in the gut microbiota of mild and severe COVID-19 patients. 

Specifically, ASVs identified as Enterococcus hirae/Enterococcus faecium, Rothia mucilaginosa, Akkermansia muciniphila, Schaalia odontolytica, Eubacterium limosum, and Slackia isoflavoniconvertens were overrepresented among patients with severe COVID-19 cases.

*Akkermansia* is a genus of Gram-negative anaerobic bacteria that belongs to the phylum *Verrucomicrobiota*. These bacteria are known for colonizing the human intestinal mucosa, have a specific ability to degrade the mucin layer, and are notably one of the most abundant single species in the human gut microbiota, accounting for approximately 0.5–5% of the total bacterial population [38]. A. muciniphila mainly degrades intestinal mucin into propionic and acetic acid. In addition, the bacterium expresses Amuc_1100, one of the most abundant pili-like proteins found on its outer membrane. Experiments on mice have shown that supplementation with *A. muciniphila* stimulates the proliferation of intestinal stem cells and enhances the differentiation of Paneth and goblet cells in the small intestine and colon in both healthy and injured mice [39]. In addition, the presence of *A. muciniphila* was associated with increased levels of acetic and propionic acids in the cecal contents of treated mice, suggesting its role in promoting intestinal mucosal repair, with short-chain fatty acids (SCFAs) playing an important role in this process [40]. The outer membrane protein Amuc_1100 has also been shown to be involved in restoring intestinal barrier function, along with acetic and propionic acids, probably by interacting with TLR2 and restoring the expression of the corresponding tight junctions [41,42].

In contrast to reports supporting the beneficial effects of *A. muciniphila*, several studies suggest potential negative effects on gut health. In mouse models of acute intestinal inflammation induced by *Salmonella enterica Typhimurium* infection and acute colitis induced by dextran sodium sulfate (DSS), *A. muciniphila* exacerbated the inflammatory response by its ability to disturb host mucus homeostasis. Simultaneously, in a study on DSS-induced colitis, it was observed that the administration of extracellular vesicles derived from *A. muciniphila* decreased the severity of DSS-induced colitis [43]. However, a later study did not confirm these findings, highlighting the need for further research to understand the complex interactions of *A. muciniphila* in gut health [44].

Our study illustrated a higher representation of A. muciniphila in the group of severe COVID-19 patients. This finding is consistent with numerous studies that have shown a significant association between a higher prevalence of the genus *Akkermansia* and SARS-CoV-2-infected subjects compared to healthy individuals [10,11,45]. Notably, researchers reported a higher abundance of *A. muciniphila* in the severe group of COVID-19 patients than in the mild group and healthy individuals [37]. An overrepresentation of *Akkermansiaceae* was detected in SARS-CoV-2-infected patients and in K18-hACE2 mice [10]. The study also observed a significant increase in mucus-producing goblet cells and a decrease in Paneth cells, specifically in the ileum of infected mice, with no such changes observed in the duodenum. The reduced Paneth cell population showed structural abnormalities, including deformed or misplaced granules, and the downregulation of several antimicrobial factors such as lysozyme, defensins, Reg3γ, and serum amyloid A in the ileum. In addition, the researchers identified a striking positive correlation between the percentage of abnormal Paneth cells and the abundance of the *Akkermansiaceae* family in the mice model [10]. Due to the controversial perspectives on the role of *Akkermansia* in human gut homeostasis, we cannot definitively conclude whether *A. muciniphila* has remodeling functions or whether it exacerbates the inflammatory response in SARS-CoV-2-infected patients. Further research and investigation are required to better understand its potential effects in these contexts.

*Enterococcus faecium* is a Gram-positive bacterium commonly found as part of the commensal flora in the human gastrointestinal tract. Despite its commensal status, it exhibits opportunistic pathogenic behavior, posing a risk to individuals with compromised immune systems or underlying health conditions. One concerning characteristic of *E. faecium* is its ability to develop resistance to many antibiotics, including those commonly used in clinical practice. The bacterium’s virulence strategy involves colonization and the secretion of various factors such as secreted antigen A, cytolysin, and gelatinase [46]. Notably, it employs specific enzymes that facilitate adherence to host tissues and inhibit the growth of competing bacteria, enhancing its ability to persist and cause infection. In addition, clinical studies have shown a remarkable relationship between the relative abundance of *Enterococcus* and certain disease parameters in SARS-CoV-2-infected patients [47]. Increased levels of *Enterococcus* have been associated with prolonged hospitalization, prolonged stays in intensive care units, increased oxygen requirements, and elevated levels of D-dimer, ferritin, and IL-6 in the bloodstream [48]. Taking into account the severity of the patient’s medical condition and the prior use of antibiotics before hospital admission, we attribute the higher prevalence of *E. faecium* in the sampled patients with severe conditions.

*Eubacterium limosum* is a producer of SCFAs, including butyrate. Increased butyrate production by this bacterium has been observed in diseases such as ulcerative colitis and experimental colitis [49]. The increase in *E. limosum* levels observed in both COVID-19 and colitis-associated gut microbiota could potentially be interpreted as an adaptive mechanism to enhance butyrate synthesis in response to a simultaneous decrease in the overall population of traditional butyrate-producing bacteria, such as *Fecalibacterium* and *Blautia*.

In patients from the mild group, we observed an enrichment of ASVs identified as Faecalibacterium prausnitzii, Ruminococcoides bili, Turicibacter sanguinis, Alistipes putredinis, Bacteroides vulgatus, Bacteroides stercoris, Lachnospiraceae bacterium sunii NSJ-8, Blautia faecis, and Anaerostipes hadrus.

Across various studies, it has been consistently observed that genera such as *Faecalibacterium*, *Blautia*, *Alistipes*, *Lachnospiraceae*, and *Bacteroides* tend to be reduced in SARS-CoV-2-infected patients and are more strongly associated with a healthy microbiome state [2,12,50]. 

In contrast to the severe group, the microbiome of mild patients exhibited an overrepresentation of flora that produces SCFAs, including *Anaerostipes hadrus*, *Faecalibacterium prausnitzii*, *Lachnospiraceae bacterium sunii NSJ-8*, and *Blautia faecis* [51]. SCFAs encompass butyrate, acetate, and propionate, which are recognized as beneficial by-products of bacterial activity that contribute to the well-being of the host. The significant role played by butyrate in constraining the expansion of opportunistic pathogens, preserving the integrity of the intestinal mucosal barrier, triggering the adaptive immune response, and reinforcing the body’s defenses against viruses is notably evident [52]. 

The positive association of *B. vulgatus*, as well as *B. stercoris* with gut flora of SARS-CoV-2-infected patients, were detected in several studies [53,54,55]. The research highlights the potential of specific *B. vulgatus* strains to positively impact the immune response, gut barrier integrity, and inflammatory processes, particularly in the context of ulcerative colitis in mice. These effects might be attributed to their ability to modulate cytokine expression, interact with colonic tissue, and influence immune cell populations [56,57]. Also, *B. vulgatus* produces multiple proteases, showing higher activity than other *Bacteroides* species. These proteases, either individually or in combination, have the potential to disturb the colonic epithelium. This disruption can result in the migration of innate immune cells, notably neutrophils, into the affected area, intensifying the inflammation associated with colitis. However, it is crucial to note that this effect was primarily observed in co-culture scenarios, as the supernatant from B. vulgatus alone did not display any adverse impact on the integrity of the colonic epithelial barrier and did not cause the disruption of the membrane integrity [58,59]. According to recent research [60] *B. stercoris* and *B. vulgatus*, in conjunction with *Prevotella copri* (which is linked to both mild and severe cases in our study), function as markers of the microbiota’s resilience to structural alterations. Interestingly, species belonging to the *Bacteroidetes* genus have been linked to the inhibition of colonic ACE2 expression, a host cell entry point for SARS-CoV-2, as demonstrated in a mouse model [61].

Through this study, we discovered an increased presence of two bile-resistant species, *Turicibacter sanguinis*, and *Ruminococcoides bili*, in the microbiome of individuals with a mild COVID-19 disease course.

The *R. bili* strains demonstrate significant resistance to bile salts, potentially facilitated by various efflux transporters that could be involved in bile export. They are also capable of metabolizing resistant starches, resulting in the synthesis of formate, lactate, and acetate. This metabolic process contributes to the well-being of other bacteria and generates beneficial SCFAs for the host [62]. *T. sanguinis* strains in the gut microbiota impact host bile and lipid compositions in a strain-specific manner. These strains possess bile salt hydrolases that influence distinct bile deconjugation patterns. Introducing Turicibacter strains led to changes in host bile acid profiles, similar to in vitro results. Mice colonized with another bacterium expressing genes from these strains exhibited reduced serum cholesterol, triglycerides, and adipose tissue mass [63].

Interestingly, we detected a higher prevalence of bacteria commonly found in the human respiratory tract. Severe patients exhibited higher levels of *Schaalia odontolytica* and *Rothia mucilaginosa*, while *Streptococcus gordonii*, *Haemophilus parainfluenzae*, and *Veillonella dispar*/*Veillonella nakazawae* were found to be more abundant in mild patients. The SARS-CoV-2 virus induces oral dysbiosis, with an increase in oral pathobionts and intestinal dysbiosis, weakening the barrier to ingested microorganisms. These conditions are critical for the successful colonization of the gut by oral pathobionts, which in turn exacerbates intestinal inflammation [7]. Among these oral pathobionts, R. mucilaginosa emerges as a potentially important factor in COVID-19. It should be considered in the diagnosis of pneumonia, regardless of the immune status of the host, because of its significant correlations with the disease [64]. Among the various bacterial taxa associated with SARS-CoV-2 infection, *R. mucilaginosa* stands out, with an increased abundance observed in both the oral and gut microbiomes [65,66].

Similarly, an increased bacterial accumulation in the gut is observed in patients with gastric achlorhydria and gastroesophageal reflux disease due to long-term proton pump inhibitor (PPI) therapy [67]. However, detailed information regarding PPI therapy is not available in our cohort.

The network analysis highlighted differences in the microbiome structure between the mild and severe groups. In the mild group, there were more microbial interactions, a higher clustering coefficient, and increased edge density, indicating a densely interconnected microbial network with efficient information transfer. In contrast, the severe group had fewer interactions, and a lower clustering coefficient and edge density, suggesting a more densely connected microbial community. These results are similar to network characteristics of respiratory microbiota from COVID-19 patients with different levels of severity [68]. These network characteristics provide insights into the organization and dynamics of microbial communities in the context of disease severity. The differences observed between the mild and severe groups suggest distinct microbial network patterns associated with different disease severities.

Besides, network analysis has been used to identify keystone taxa (hubs) within the microbiomes of COVID-19 patients. These keystone taxa are specific microorganisms that have been found to have a significant impact on the overall structure and composition of the microbial community within COVID-19 patients. 

In cases of a mild course of infection, three taxa, including *Dorea formicigenerans*, *Blautia obeum*, and *Coprococcus comes*, came to the forefront as keystone species. *D. formicigenerans* belongs to the *Lachnospiraceae* family and is known for its prolific production of formic acid. Research has demonstrated that the administration of formic acid to pigs leads to elevated levels of beneficial microorganisms while concurrently inhibiting the growth of pathogenic members of the Enterobacteriaceae family [69]. Notably, *B. obeum* and *C. comes* have previously been reported to be associated with a mild/moderate COVID-19 disease course and are positively correlated with lymphoid-related markers, suggesting a possible interaction between these gut microbes and the regulation of lymphocytes [37].

In the severe group, two ASVs that belong to *Lachnospiraceae bacterium* and Jingyaoa shaoxingensis should be considered as taxa hubs. The role of *L. bacterium* as a keystone species in the microbiome of severe patients may indicate an adaptive response to the reduction in the overall population of conventional butyrate-producing bacteria, similar to *E. limosum*.

Our study contributes to the spectrum of publications analyzing the composition of the gut microbiome in patients with COVID-19, expanding the geography of research to Russia. In our study, we aimed to understand the contribution of gut microbiome composition to the severity of COVID-19, in line with other inflammatory and infectious diseases which have been described previously [70].

While under quarantine conditions due to the COVID-19 pandemic, we limited our sample collection to a single hospital in accordance with the research objectives. We deliberately chose not to include a control group of patients, as mentioned above. In addition, patients who were admitted to the hospital in critical condition were not included in the study group due to the inability to collect biomaterial during resuscitation efforts. Despite these limitations, our analysis of samples from a fully characterized group of 92 COVID-19 patients is sufficient to draw reliable conclusions.

Taken together, we can conclude that the gut microbiota, as well as the upper respiratory tract microbiota [30], of mild COVID-19 patients is very similar to that of healthy individuals. This trend can be observed both at the level of taxonomic composition and at the level of microbial interaction network density. The lack of major perturbations in the microbiome of these patients may be both a cause and a consequence of the milder disease course. We cannot answer this question in this study.

In the group of severe patients, we observe a replacement of the main butyrate-producing bacteria such as *Fecalibacterium* and *Blautia* by *E. limosum*, which could be interpreted as an adaptive mechanism to enhance butyrate synthesis. This assumption holds for the switch of keystone taxa from *B. obeum* in mild patients to *L. bacterium* in severe groups that we observed using network analysis, highlighting the key role of butyrate-producing bacteria for the gut microbiota.

In summary, our findings suggest that the clinical manifestation of SARS-CoV-2 infection has different effects on the composition of the gut microbial community in both mild and severe groups of patients, suggesting differential microbial responses. While the current findings are derived from retrospective data, they lay the groundwork for prospective studies to validate the gut microbiome’s role as a predictive biomarker for COVID-19 severity.

## 5. Limitations

Findings from the present study should be interpreted in light of its limitations, which include a relatively small study cohort. While under quarantine conditions due to the COVID-19 pandemic, we limited our sample collection to a single hospital. In addition, patients who were admitted to the hospital in critical condition were not included in the study group due to the inability to collect biomaterial during resuscitation efforts.

The lack of a control group (patients without COVID-19) in our study was primarily due to the fact that the research was conducted in a hospital that had been converted specifically for the treatment of COVID-19 patients. To address the potential impact of pre-existing conditions on microbiome composition, we carefully excluded patients with oncologic conditions, and other chronic conditions known to significantly alter the microbiota. This decision was guided by the goal of minimizing confounding variables and simulating a ‘healthy’ microbiota baseline as closely as possible. Thus, our study cohort can be considered as having a relatively uniform health baseline prior to COVID-19 infection. Ideally, the control group for our study would have consisted of the same 92 patients prior to their SARS-CoV-2 infection. However, due to the nature and timing of the pandemic, it was not possible to obtain pre-infection baseline data for these patients. This limitation underscores the inherent challenges of conducting retrospective microbial research during acute public health crises, when pre-infection data are rarely available.

Since antibiotics can be obtained without a doctor’s prescription in Russia, and our knowledge of pre-hospital antibiotic usage relies on patients’ self-disclosures, obtaining accurate information about the particular antibiotics used poses a challenge. Nevertheless, we cannot ignore this information, as it may have a significant impact on the microbiome.

While the study provides insights into how the gut microbiota composition correlates with disease severity, it stops short of establishing a direct causal relationship between specific microbiota changes and COVID-19 outcomes due to vertical analysis.

Despite these limitations, our analysis of samples from a fully characterized group of 92 COVID-19 patients is sufficient to draw reliable conclusions.

## Figures and Tables

**Figure 1 biomedicines-12-00996-f001:**
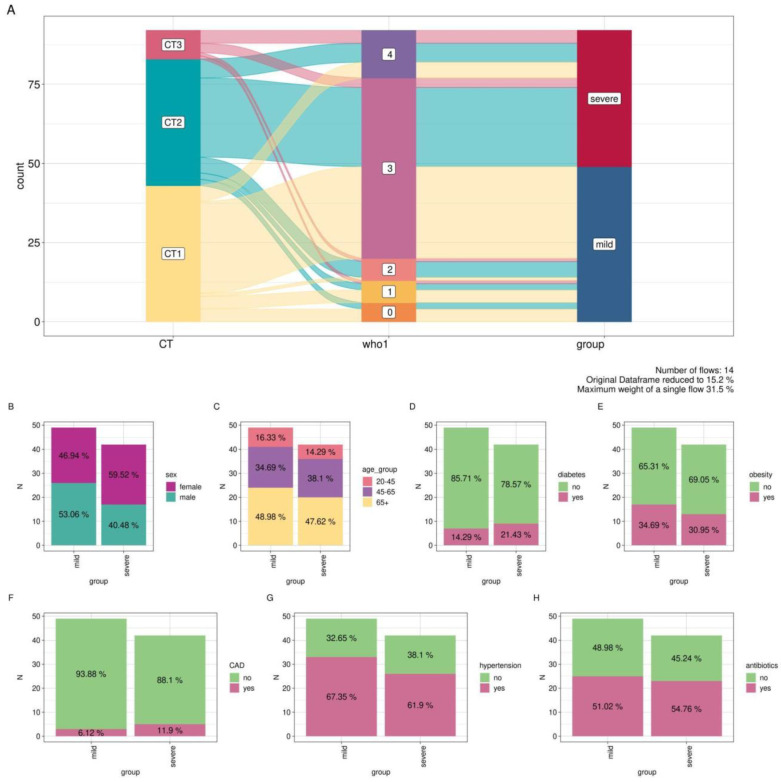
Study cohort overview. (**A**) Alluvial plot of comparison groups based on CT and WHO-PS; (**B**–**H**) distribution of metadata by comparison group (N-number of counts).

**Figure 2 biomedicines-12-00996-f002:**
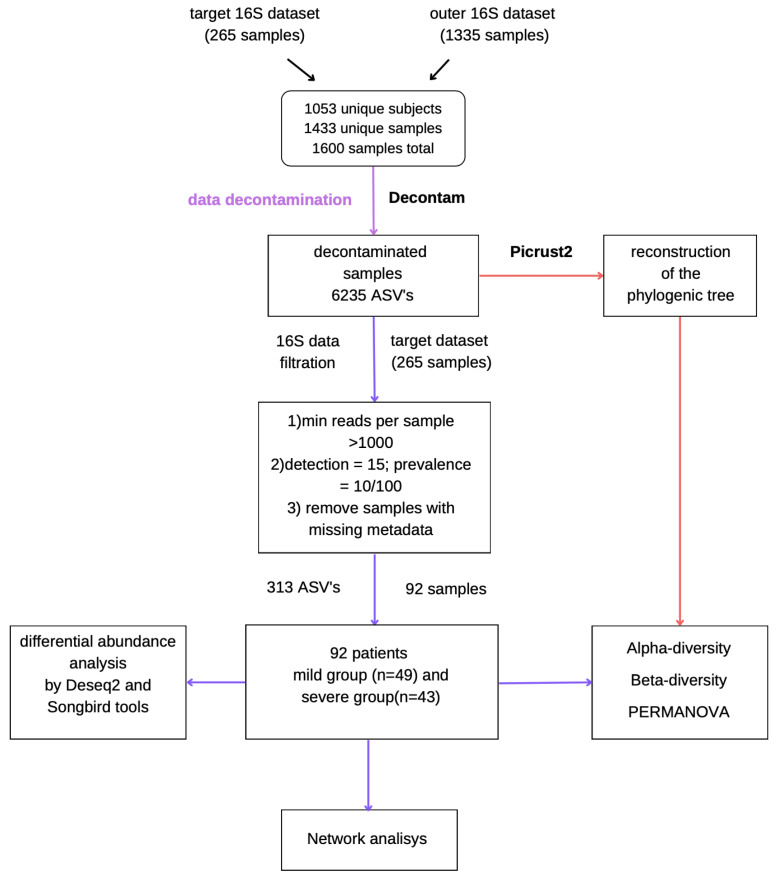
Data preprocessing scheme.

**Figure 3 biomedicines-12-00996-f003:**
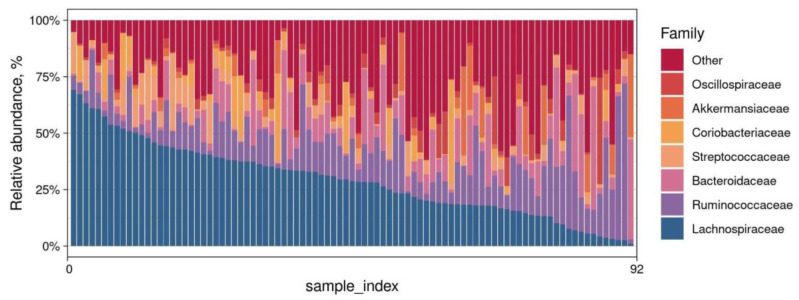
Relative abundance (%) of the 7 most abundant families found in the samples.

**Figure 4 biomedicines-12-00996-f004:**
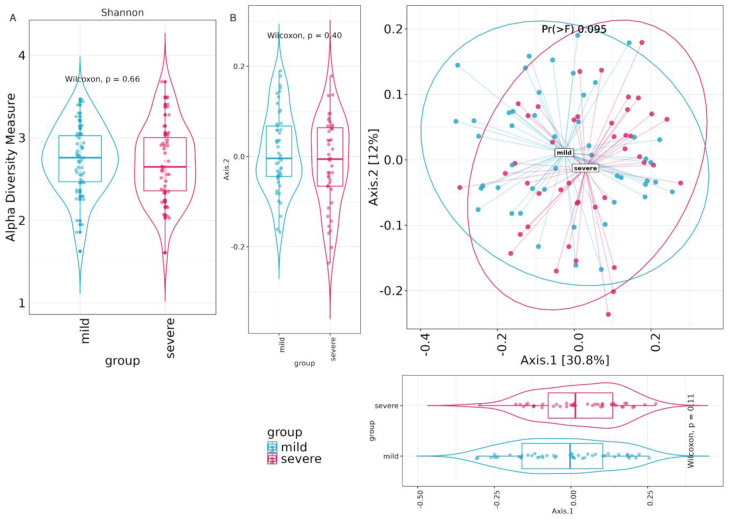
(**A**) Alpha diversity by Shannon index. Box plots illustrate alpha diversity by Shannon index in bacterial microbiomes of 92 patient samples from mild (*n* = 49) and severe (*n* = 43) groups and statistical test results (Wilcoxon test). Median values and interquartile ranges have been indicated in the plots. (**B**) Principal coordinate analysis (PCoA). PCoA plots of beta-diversity estimates (weighted Unifrac) and statistical test results (Adonis test) in the mild and severe group.

**Figure 5 biomedicines-12-00996-f005:**
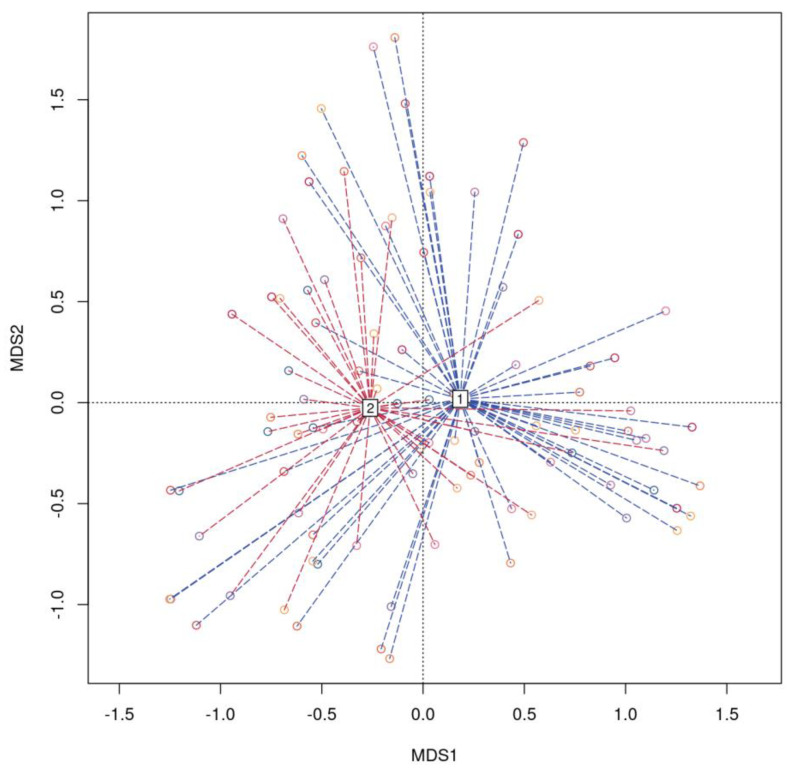
Canonical analysis of principal coordinates (CAP). CAP plot representing the Bray-Curtis dissimilarities between samples that belong to fecal community type 1 (1, shown in red) and 2 (2, shown in blue).

**Figure 6 biomedicines-12-00996-f006:**
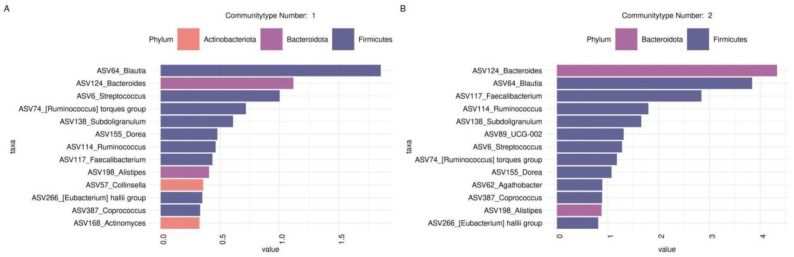
Barplots illustrate top drivers contributing to each community type identified with DMM. (**A**) Top taxonomic drivers (at ASV level ) community number 1; (**B**) Top taxonomic drivers (at ASV level ) community number 2.

**Figure 7 biomedicines-12-00996-f007:**
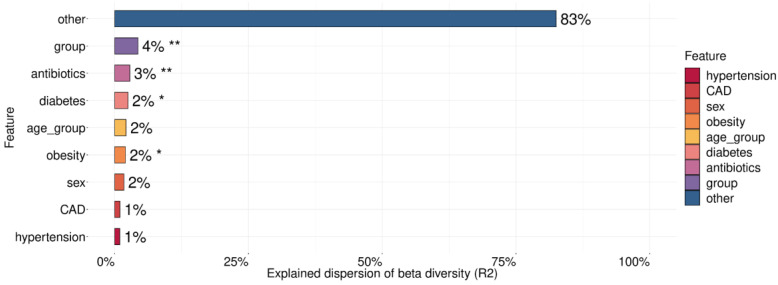
Metadata factors explain the variance in microbiome composition. Asterisks indicate the level of significance with the following thresholds: (*) = 0.05, (**) = 0.01.

**Figure 8 biomedicines-12-00996-f008:**
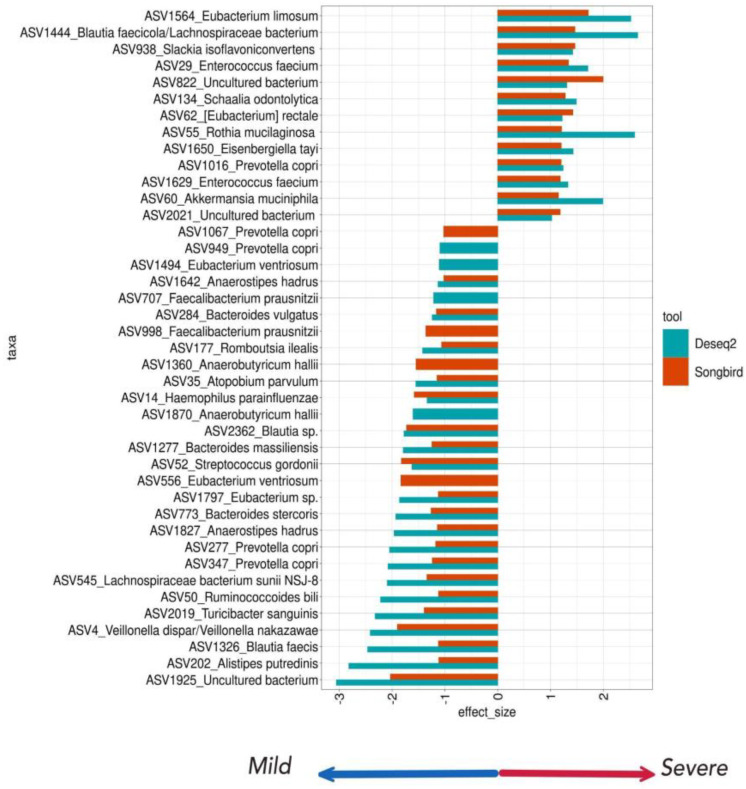
The Deseq2 and Songbird utilities analyzed the differentially abundant taxa testing associations. ASVs associated with a milder course of COVID-19 have a lower effect size (blue arrow); ASVs associated with a more severe course of COVID-19 have a larger effect size (red arrow). The tool is marked in color. Taxa names consist of ASV numbers and species according to BLASTN alignment.

## Data Availability

Sequencing reads for 16S rRNA gene amplicons were deposited to NCBI BioProject under project name PRJNA989180. https://www.ncbi.nlm.nih.gov/bioproject/PRJNA989180 (accessed on 29 June 2023).

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
