# Peer review of "Microbial Signatures in COVID-19: Distinguishing Mild and Severe Disease via Gut Microbiota"

_biomedicines, 2024, doi:10.3390/biomedicines12050996_

Round 1
Reviewer 1 Report
Comments and Suggestions for Authors
This is a significant study in an important matter: Distinguishing Mild and Severe Disease of COVID-19 via Gut Microbiota.
The work is well outlined, and well organized. The experiments are well organized and the results support the conclusion. Overall, it is a significant study on this matter. However, to be published, in my opinion, there are some major and minor changes to be made:
1) Page 6, Line 24-233: This part belongs to the Materials and methods, not results.
2) Page 3, Lines 102-118: Choose between dots and – (not both).
3) Page 5, Line 207: The paragraph “This section may be divided by subheadings. It should provide a concise and precise description of the experimental results, their interpretation, as well as the experimental conclusions that can be drawn.” Is not necessary. This is the what is supposed to be always in the “Results section”.
4) The purpose of the manuscript must be clearly explained in the abstract and then in the conclusions and to what extent it may have clinical implications. In other words, to what extent can the analysis of the microbiota be used as a predictor of COVID outcome, or if it can only be done retrospectively.
Author Response
We sincerely appreciate your time and effort in reviewing this manuscript.
1) Page 6, Line 24-233: This part belongs to the Materials and methods, not results.
In the new version of this article, we moved this section to materials and methods.
2) Page 3, Lines 102-118: Choose between dots and – (not both).
In the revised version of this article, we have corrected this issue.
3) Page 5, Line 207: The paragraph “This section may be divided by subheadings. It should provide a concise and precise description of the experimental results, their interpretation, as well as the experimental conclusions that can be drawn.” Is not necessary. This is the what is supposed to be always in the “Results section”.
In the revised version of this article, we have corrected this issue.
4) The purpose of the manuscript must be clearly explained in the abstract and then in the conclusions and to what extent it may have clinical implications. In other words, to what extent can the analysis of the microbiota be used as a predictor of COVID outcome, or if it can only be done retrospectively.
While the current findings are derived from retrospective data, they lay the groundwork for prospective studies to validate the gut microbiome's role as a predictive biomarker for COVID-19 severity. Future research could then integrate these insights into clinical practice, utilizing the gut microbiome for risk stratification and more personalized patient management. We have added this to the new version of the article
Reviewer 2 Report
Comments and Suggestions for Authors
The manuscript represents a study of faecal microbiota in SARS-CoV-2 infected individuals. The rationale and methodology are adequate despite the low number of samples. However, some issues should be clarified. Figure 2 is misleading, and I suggest transforming parts B to F into a table in which the main columns are divided by gender and age and then each condition, hypertensive, BMI, and antibiotic therapy. Figure 3 is important, but it has to be related to the table; certain microbiota changes may not be specific to COVID-19 but seem more appropriate for hypertensive patients—figure 4 part B. The axis 1 -0.4 to -0.2 is informative; most samples are mild, and only one is severe. Thus, the primary component analysis has to be performed with other parameters. Two questions have to be solved: strain of the virus and exposure, treatment for comorbidities and COVID-19 characteristics of the patients. Were there other treatments? Any other relevant medical history? Vaccination? The manuscript could benefit from a reduction in the discussion and the inclusion of limitations of the study. One of the limitations is a vertical analysis which should be discussed in light of the results.
Comments on the Quality of English LanguageSeveral grammatical mistakes were encountered.
Author Response
We sincerely appreciate your time and effort in reviewing this manuscript.
Figure 2 is misleading, and I suggest transforming parts B to F into a table in which the main columns are divided by gender and age and then each condition, hypertensive, BMI, and antibiotic therapy.
In Figure 2 (panels B-F), we aimed to illustrate the distribution of traits across comparison groups, thereby extending the validity of comparisons among the groups rather than focusing on individual patient characteristics. Additionally, we have included a table detailing patient characteristics in the supplementary materials to provide a comprehensive overview.
Figure 3 is important, but it has to be related to the table;
We have included a table of relative abundances in the revised version of the manuscript to provide detailed insights into the data.
certain microbiota changes may not be specific to COVID-19 but seem more appropriate for hypertensive patients—figure 4 part B.
We acknowledge the impact of comorbidities and validate that the groups were statistically indistinguishable based on these parameters.
In addition, the PERMANOVA analysis (Figure 7), showed that the COVID-19 severity group contributed significantly more variation to the data (4% **) highlighting the influence of disease severity on study outcomes, where hypertension (1% and not significant). We also accounted for these parameters ( age, sex, hypertension, diabetes, obesity, and coronary artery disease (CAD) ) in the differential abundance analysis by including these variables as covariates.
The axis 1 -0.4 to -0.2 is informative; most samples are mild, and only one is severe. Thus, the primary component analysis has to be performed with other parameters.
we have changed this picture in the new version of the article
Two questions have to be solved: strain of the virus and exposure, treatment for comorbidities and COVID-19 characteristics of the patients. Were there other treatments?COVID-19 characteristics of the patients.Any other relevant medical history?
Vaccination?
In this study, although we confirmed the presence of virus in patients, sequencing the virus to determine its strain was not our goal. During the sample collection period from April 15 to June 27, 2021, the Delta variant (B.1.617.2) predominated in Russia. Patients were usually hospitalized 5-7 days after the onset of symptoms as soon as complications requiring medical monitoring developed. All patients were unvaccinated. Stool samples were collected on admission to the hospital, prior to COVID-19 treatment, which did not take into account comorbidities. The treatment regimen included glucocorticosteroids (dexamethasone), mucolytic agents (acetylcysteine), anticoagulants (enoxaparin, heparin), non-steroidal anti-inflammatory drugs (paracetamol, ibuprofen) and other symptomatic therapy. In case of bacterial infections, antibiotics (penicillins, cephalosporins, macrolides) were prescribed. Additional treatment included antiviral agents (remdesivir), antihypertensive drugs (β-blockers, ACE inhibitors), antidiabetic drugs (short- and long-acting insulins) and probiotics. It is worth emphasizing again that although treatment was given, fecal samples were taken prior to the time of treatment. COVID-19 patients in this study were categorized based on the severity of their infection, ranging from the 1st to the 3rd degree of pneumonia severity as per CT scans and the WHO scale criteria for assessing severity (Figure 2 A). Treatment protocols and the interpretation of clinical data adhered to the then-current interim guidelines issued by the Ministry of Health of Russia (version 10, dated 08.02.2021).
The manuscript could benefit from a reduction in the discussion and the inclusion of limitations of the study. One of the limitations is a vertical analysis which should be discussed in light of the results.
We have added a limitations section to the new version of the article.
Reviewer 3 Report
Comments and Suggestions for Authors
Dear authors,
I have now completed the review of the manuscript titled "Microbial Signatures in COVID-19: Distinguishing Mild and Severe Disease via Gut Microbiota."
The manuscript is interesting and, in general, fairly well-written.
I have some suggestions to further improve the quality of the manuscript.
I would like to suggest that the authors address these limitations in the article, either by discussing them in the limitations section or, where feasible, by making the appropriate revisions:
1. The sample size of 92 patients, while not insignificant, is still relatively small to draw broad conclusions, especially when splitting into mild and severe groups. A larger sample size would provide more confidence in the results.
2. There is no healthy control group for comparison. While the authors justify this by saying they wanted to focus on disease severity, including healthy controls could provide valuable context to interpret the microbiome differences between mild and severe COVID-19. However, I think this can be justified using previous research. For example, in Immunogenicity of COVID-19 Vaccines in Patients with Diverse Health Conditions: a Comprehensive Systematic Review, etc...
3. The information on patients' prior antibiotic usage relies on self-reporting, which may not be fully reliable. More controlled data on antibiotic exposures would strengthen the analysis.
4. The categorization of patients into mild vs severe groups, while reasoned, still involves some subjectivity in combining CT scores and WHO scales. A more standardized severity classification system would allow better comparisons to other studies.
5. Associations between gut microbiome composition and COVID-19 severity are reported, but the cross-sectional nature of the sampling makes it difficult to establish causal relationships. Longitudinal sampling could help clarify directionality of host-microbiome interactions. Also, authors should look into COVID-19 susceptibility and clinical outcomes in inflammatory bowel disease: An updated systematic review and meta-analysis. Since the gut microbiome in COVID-19 related to COVID-19 susceptibility and outcomes in inflammatory bowel disease, which also involves gut health, could offer useful comparative insights.
6. More discussion of potential confounding factors like diet, medications, comorbidities etc would be helpful to contextualize the observed microbiome associations with COVID-19 severity. For example, See the An Adaptive Ensemble Deep Learning Framework for Reliable Detection of Pandemic Patients. This article discusses using deep learning models to detect pandemic patients, which could be useful for developing computational methods to analyze gut microbiome data in the context of COVID-19 severity prediction.
7. The functional implications of the microbiome differences could be explored in more depth, beyond taxonomic associations. Metagenomic sequencing or metabolomics could provide further mechanistic insights.
Thank you for your valuable contributions to our field of research. I look forward to receiving the revised manuscript.
Author Response
We sincerely appreciate your time and effort in reviewing this manuscript.
1.The sample size of 92 patients, while not insignificant, is still relatively small to draw broad conclusions, especially when splitting into mild and severe groups. A larger sample size would provide more confidence in the results.
While our sample size is relatively limited, several factors contributed to this result. Transfers of patients to intensive care units or other specialized hospitals frequently disrupted the collection process, complicating our ability to collect consistent biomaterials. Our analysis focused specifically on fecal samples from COVID-19 patients for whom comprehensive metadata was available, including age, sex, CT lung scans, WHO performance status (WHO-PS), antibiotic use, and the presence of comorbidities such as hypertension, coronary artery disease, diabetes, and obesity. This rigorous requirement for complete metadata necessarily reduced our sample size but allowed for more robust statistical analysis. Of the 265 inpatients initially enrolled in COVID-19, only 92 samples met the criteria for inclusion in the final analysis. Although this approach limited the number of samples, it significantly improved the quality and reliability of our data, providing a solid foundation for meaningful conclusions.
- There is no healthy control group for comparison. While the authors justify this by saying they wanted to focus on disease severity, including healthy controls could provide valuable context to interpret the microbiome differences between mild and severe COVID-19. However, I think this can be justified using previous research. For example, in Immunogenicity of COVID-19 Vaccines in Patients with Diverse Health Conditions: a Comprehensive Systematic Review, etc...
The lack of a control group (patients without COVID-19) in our study was primarily because the research was conducted in a hospital that had been converted specifically for the treatment of COVID-19 patients. To address the potential impact of pre-existing conditions on microbiome composition, we carefully excluded patients with oncologic conditions, inflammatory bowel disease, and other chronic conditions known to significantly alter the microbiota. This decision was guided by the goal of minimizing confounding variables and simulating a baseline 'healthy' microbiota as closely as possible. Thus, our study cohort can be considered as having a relatively uniform health baseline before COVID-19 infection. Ideally, the control group for our study would have consisted of the same 92 patients before their SARS-CoV-2 infection. However, due to the nature and timing of the pandemic, it was not possible to obtain pre-infection baseline data for these patients. This limitation underscores the inherent challenges of conducting retrospective microbial research during acute public health crises when pre-infection data are rarely available.
- The information on patients' prior antibiotic usage relies on self-reporting, which may not be fully reliable. More controlled data on antibiotic exposures would strengthen the analysis.
During the pandemic, the hospital was temporarily repurposed for inpatient Covid-19 treatment. Since patients from all over the city were admitted, it was not possible to obtain medical records from other hospitals, and information on antibiotic use was obtained in self-report format.
Additionally, the PERMANOVA analysis (Figure 7) indicated that the intake of antibiotics significantly contributed to the variation in the data (3% **), underscoring the impact of this factor, as anticipated from antibiotic use.
We added this aspect to the limitation section in the new version of the paper.
- The categorization of patients into mild vs severe groups, while reasoned, still involves some subjectivity in combining CT scores and WHO scales. A more standardized severity classification system would allow better comparisons to other studies.
The criteria for assessing severity, treatment protocols, and the interpretation of clinical data were guided by the interim guidelines issued by the Ministry of Health of Russia at the time (version 10, dated February 8, 2021). Consequently, our assessment is constrained by the applicable severity data. However, we recognize the need for a more standardized method to evaluate the severity of COVID-19 in patients.
- Associations between gut microbiome composition and COVID-19 severity are reported, but the cross-sectional nature of the sampling makes it difficult to establish causal relationships. Longitudinal sampling could help clarify directionality of host-microbiome interactions. Also, authors should look into COVID-19 susceptibility and clinical outcomes in inflammatory bowel disease: An updated systematic review and meta-analysis. Since the gut microbiome in COVID-19 related to COVID-19 susceptibility and outcomes in inflammatory bowel disease, which also involves gut health, could offer useful comparative insights.
Thanks for the useful links to help improve our article, we have added this aspect to the discussion section line 418-426 in the new version of the paper
- More discussion of potential confounding factors like diet, medications, comorbidities etc would be helpful to contextualize the observed microbiome associations with COVID-19 severity. For example, See the An Adaptive Ensemble Deep Learning Framework for Reliable Detection of Pandemic Patients. This article discusses using deep learning models to detect pandemic patients, which could be useful for developing computational methods to analyze gut microbiome data in the context of COVID-19 severity prediction.
Thanks for the useful links to help improve our article, we have added this aspect to the discussion section line 418-426 in the new version of the paper.
- The functional implications of the microbiome differences could be explored in more depth, beyond taxonomic associations. Metagenomic sequencing or metabolomics could provide further mechanistic insights.
We agree that exploring the functional implications of microbiome differences through methods like metagenomic sequencing or metabolomics could enhance the analysis by providing deeper mechanistic insights. However, given the retrospective nature of our research, it was not feasible to perform these analyses within the current study's framework.
Round 2
Reviewer 2 Report
Comments and Suggestions for Authors
The authors responded to most of the queries raised. In my opinion, the manuscript was improved and can be published.
Comments on the Quality of English LanguageSeveral grammatical mistakes were enocuntered.
Reviewer 3 Report
Comments and Suggestions for Authors
All comments have been thoroughly addressed. I extend my gratitude to both the authors and editors for taking my opinions into consideration during the review of this manuscript.